# WIFF1.0: A hybrid machine-learning-based parameterization of Wave-Induced sea-ice Floe Fracture.

Christopher Horvat[a] and Lettie A. Roach[b,c]

[a]Institute at Brown for Environment and Society, Brown University, Providence, RI, USA
[b]Department of Atmospheric Sciences, University of Washington, Seattle, WA, USA
[c]NASA Goddard Institute for Space Studies and Center for Climate Systems Research, Columbia University, New York, NY, USA

**Correspondence:** Christopher Horvat (horvat@brown.edu)

**Abstract.** Ocean surface waves play an important role in maintaining the marginal ice zone, a heterogenous region occupied by sea ice floes with variable horizontal sizes. The location, width, and evolution of the marginal ice zone is determined by the mutual interaction of ocean waves and floes, as waves propagate into the ice, bend it, and fracture it. In previous work, we developed a one-dimensional "superparameterized" scheme to simulate the interaction between the stochastic ocean surface wave field and sea ice. As this method is computationally expensive and not bitwise reproducible, here we use a pair of neural networks to accelerate this parameterization, delivering an adaptable, computationally-inexpensive, reproducible approach for simulating stochastic wave-ice interactions. Implemented in the sea ice model CICE, this accelerated code reproduces global statistics resulting from the full wave fracture code without increasing computational overheads. The combined model, Wave-Induced Floe Fracture (WIFF v1.0) is publicly available and may be incorporated into climate models that seek to represent the effect of waves fracturing sea ice.

## 1 Introduction

Sea ice is a composite of individual pieces, called floes. Floes are discrete solid pieces of sea ice with horizontal geometric length scales (sizes) ranging from meters to tens of kilometers. Yet in climate model simulations, sea ice is treated as a continuum field (Hibler, 1979; Golden et al., 2020). To represent the effect of changes to sea ice floe geometry in climate models, modeling centers have started to incorporate prognostic sub-grid-scale parameterizations of the sea ice floe size distribution (FSD) as a core aspect of their model physics (Horvat and Tziperman, 2015; Zhang et al., 2015; Roach et al., 2018b; Boutin et al., 2018; Bateson et al., 2020; Hunke et al., 2019). The FSD is a probability distribution $f(r)$, where $f(r)dr$ is the percentage of a grid area comprised of floes with a size between $r$ and $r + dr$. The integral of $f(r)$ over all positive $r$ is equal to the sea ice concentration, $c$.

One major advantage of simulating a sub-grid-scale FSD is it permits the representation of coupled interactions between sea ice floes and ocean surface waves. Ocean surface waves propagate into, are attenuated by, and fracture sea ice in regions known as the marginal ice zone (MIZ, Wadhams et al., 1988; Langhorne et al., 1998; Squire et al., 1995). Wave-affected sea-ice-covered regions are observed to be several million square kilometers in size in both hemispheres, impacting up to half of the

sea ice cover depending on the season and hemisphere (Horvat et al., 2020). When waves propagate into and fracture sea ice, there is no direct change to sea ice concentration or thickness. Instead, this fracture process alters sea ice floe sizes, increasing the sensitivity of the sea ice to external thermodynamic or dynamic forcing (Steele, 1992; Feltham et al., 2006; Horvat et al., 2016; Horvat and Tziperman, 2017) and altering the attenuation of ocean wave energy (Perrie and Hu, 1996; Meylan et al., 2021). Thus there is a hypothesized coupling between sea ice loss and wave activity - waves propagate into the ice, fracturing it, causing it to melt, and enhancing wave propagation (Kohout et al., 2011; Asplin et al., 2012, 2014). Floe fracture by waves is the dominant process driving changes in floe perimeter in summer months when the most lateral melting occurs (Roach et al., 2019).

In a climate model, coupled wave-ice feedbacks are related to two sub-grid-scale distributions: the FSD, $f(r)$, and the ocean surface wave spectrum, $S(\lambda)$, where $\lambda$ is the wavelength and $\int S(\lambda)d\lambda = E$ (units m$^2$) is the wave energy per square meter, with $E = 4\sqrt{H_s}$ for $H_s$ the significant wave height (Michel, 1968). Heuristic parameterizations have been developed to relate bulk properties of the ocean surface wave field to the FSD - generally assuming that fractured floes follow a power-law distribution (Williams et al., 2013; Zhang et al., 2015; Bateson et al., 2020). Yet there is conflicting evidence about whether power-law FSDs are observed in nature (Herman, 2013; Stern et al., 2018a, b; Horvat et al., 2019), and whether power-law size distributions are generated by the process of wave-induced floe fracture (Horvat et al., 2016; Herman et al., 2021).

One challenge in relating $S(\lambda)$ and $f(r)$ is that the response of sea ice is determined by the two-dimensional ocean height field, a stochastic (i.e. one of many possible) representation of the ocean wave spectrum. An approach designed by Horvat and Tziperman (2015) was to generate a high-resolution, one-dimensional ocean surface wave field in every ice-covered grid cell, and explicitly resolve the strain experienced by the ice, the attenuation of wave energy, and the resulting statistics of sea ice fracture. This method (hereafter called SP-WIFF) is broadly analogous to a "super-parameterization" (SP) approach used in coupled climate models for parameterization of cloud effects (Randall et al., 2003; Grabowski, 2004) or oceanic deep convection (Campin et al., 2011). SP-WIFF is included as a component of the sea ice model CICE (Roach et al., 2018b; Hunke et al., 2019), and has been coupled to an ocean surface wave model (Roach et al., 2019). Running SP-WIFF incurs high computational costs - in the case of its incorporation in CICE, it increases computation times by an order of magnitude (see Sec. 4). As it is stochastic SP-WIFF is not bitwise reproducible: two identically initialized and forced simulations using SP-WIFF will not produce identical output up to the level of machine precision, which is often a necessary feature for climate model development.

To provide a computationally inexpensive, flexible, and bit-wise reproducible sub-grid-scale parameterization of wave-induced ice fracture, we here present an accelerated parameterization of wave-induced floe fracture using a pair of neural networks for input classification and sea ice fracture (NN-WIFF). The full code, WIFF1.0 (Wave-Induced Floe Fracture), is publicly available, and contains SP-WIFF and NN-WIFF, along with code for retraining or adapting NN-WIFF to prescribed error thresholds, variable input/output variables, and model configurations. The model is trained using 5.1 million input and output vectors taken from coupled CICE-WAVEWATCH 3 simulations (see Sec. 3). Implemented in free-running coupled simulations in a year not corresponding to the training dataset, NN-WIFF produces global sea ice variability that is not statistically significantly different from those produced by SP-WIFF, while reducing its computational overhead by more than 95%.

## 2 SP-WIFF: a super-parameterized wave fracture scheme

The super-parameterization of sea ice fracture by ocean waves was derived in Horvat and Tziperman (2015) (Sec. 2.3), its statistical properties explored in Horvat and Tziperman (2017) (Sec. 3.4), was implemented with offline wave-ice interactions in a sea ice model in Roach et al. (2018b) (Sec. 2.4), and was introduced into a fully-coupled wave-ice model in Roach et al. (2019). We summarize it here.

Consider a region corresponding to a climate model grid cell where ocean surface wave energetics are described by a discrete uni-directional wave spectrum:

$$S(\lambda_i)d\lambda_i = \int\limits_0^{2\pi} S(\lambda_i,\Theta)d\lambda_i d\Theta. \tag{1}$$

Considering only those floes with horizontal size between $r$ and $r+dr$, a fraction of the domain $\frac{f(r)}{\tau}\Omega(r,t)\,dr\,dt$ (unitless) is broken by ocean surface waves over a period $dt$. The parameter $\tau$ is a prescribed timescale over which the floe fracture takes place. In the Roach et al. (2018a) implementation, $\tau$ is determined via an adapting timestepping algorithm to satisfy the CFL criteria for fast-propagating waves and small-area FSD categories, respectively (see Horvat and Tziperman (2017), Appendix A for further details). The fractured area has its own floe size distribution - the fraction of $\Omega(r,t)dr\,dt$ that now belongs to floes with size between $s$ and $s+ds$ is $F(r,s)\,ds$ (unitless), with $\int\limits_0^\infty F(r,s)ds = 1$. Generically, the time rate of change of area of floes of size $r$ due to fracture by ocean surface waves is,

$$\frac{\partial f(r,t)}{\partial t} = \frac{1}{\tau}\left[ -f(r,t)\Omega(r,t) + \int\limits_r^\infty \Omega(s,t)f(s,t)F(s,r,t)\,ds \right]. \tag{2}$$

Note that for ease of interpretation, notation in Eq. 2 differs from the analogous equation set in Horvat and Tziperman (2015), Eq.s 19-23. Equation 2 is one tendency term in evolution of the FSD, which responds to multiple external forcings (e.g., thermodynamic growth/melting and advection) as described in Horvat and Tziperman (2015). External forcing terms are computed at each model timestep, but the FSD $f(r,t)$ is prognostically evolved using the above-referenced adaptive timestepping scheme. Wave-ice coupling is performed at each sea ice model grid step, allowing for feedbacks between the FSD and, for example, floe-size-dependent wave attenuation schemes (Meylan et al., 2021).

The first term in Eq. 2 is the loss of floe area at size category $r$ per unit time, and the second is the increase in floe area in size category $r$ due to the fracture of floes of larger sizes. The limits of integration reflect the fact that any given floe can not fracture into a larger floe, and therefore $F(s,r) = 0$ for $s < r$. As Eq. 2 is a generic tendency equation for fractured sea ice, SP-WIFF, then, refers to a parameterization of both $\Omega$ and $F$, which in each ice thickness category are evaluated as follows:

**S1:** The discrete one-dimensional wave spectrum $S(\lambda)d\lambda$ is converted to a 1-dimensional ice strain field $\eta(x)$ of (arbitrarily chosen) length 10 km, (see (Horvat and Tziperman, 2015, eq. 20-21))

**S2:** A collection of fracture lengths $\{L\}$ is found by finding the distance between successive strain extrema that would fracture a floe. These are binned into a probability distribution $A(r)$, where $A(r)dr$ is the length-weighted percentage of

fracture lengths with length between $r$ and $r + dr$. Note we have replaced $r \cdot R(r)$ in Horvat and Tziperman (2015) with $A(r)$ here to simplify notation.

**S3: S1-S2** are repeated (increasing the size of $\{L\}$) until convergence, defined as a mean absolute difference between successively updated values of $A(r)$ below $5 \times 10^{-4}$.

Each wave field and corresponding fracture length collection (steps **S1** and **S2**) is performed independently, i.e. on an unbroken floe of length 10km. We next use the histogram $A(r)dr$ to compute $\Omega$ and $F$. First,

$$\Omega(r,t) = \int_0^r A(s,t)ds, \tag{3}$$

and $\Omega(r,t)\,dr$ is equal to the length-weighted fraction of all fracture lengths smaller than $r$, which assuming a random horizontal distribution of floes is also equal to the fraction of floes of size $r$ that will be broken. The resulting distribution of floe sizes is determined by the histogram itself,

$$F(r,s,t)ds = \begin{cases} A(s,t)ds / \int_0^r A(l,t)dl = A(s,t)ds/\Omega(r,t) & \text{if } r \geq s, \\ 0 & \text{if } r < s. \end{cases} \tag{4}$$

The denominator in Eq. 4 comes to ensure normalization of $F(r,s,t)$ over $s$, and therefore $F(r,s,t)ds$ is the probability distribution of floe sizes from a broken floe of size $r$.

SP-WIFF therefore involves a stochastic component (**S1**) and an expensive peak-finding and histogram-generation component (**S2**). Its execution time is uncertain because the number of steps until convergence (**S3**) is not known in advance. Note that this convergence is related to the stochastic histogram generation code, which converges to a steady-state distribution of fractures, and is not related to feedbacks from the waves to the ice. Previously, to circumvent these challenges, SP-WIFF was implemented in CICE using a fixed phase in step **S1** and only iterated once in step **S3**. We refer to this single-iteration implementation as SP-WIFF$_1$. Despite this simplification, CICE runtimes are increased by approximately a factor of 4 (see Sec. 4) over benchmark estimates without wave fracture. We seek to improve the performance of SP-WIFF by replacing **S1-S3** with a neural network trained on SP-WIFF input and output data, which we call NN-WIFF.

## 3 Accelerating the parameterization of coupled ice-wave interactions

The NN-WIFF code replaces the full super-parameterization (SP-WIFF) with a parameterization that replicates its statistics but has reduced computational overhead. It consists of two full-connected feedforward neural networks trained using the Python Keras deep learning library, one 100 node by 100 node input classification scheme for determining whether NN-WIFF will run, and a second with five hidden layers of 100 nodes each, for generating fracture histograms. We introduced a classifier layer as SP-WIFF frequently returns un-fractured sea ice in low-wave regimes, and we wish to train and run the histogram-generating network only when the sea ice will fracture. Recent observations of universal threshold behavior for sea ice break-up

(Voermans et al., 2020) may provide the opportunity in future work to replace this classification layer with a simple physically-based threshold for when the sea ice fractures.

Network architectures were chosen using a metalearning approach of varying loss functions and numbers of nodes and layers (we provide code for altering network size in the WIFF1.0 release). Training is performed using the 'adam' optimization scheme using the resilient backpropagation method. Rectified linear unit (RELU) activation functions are used for each of the hidden neurons, and we use a softmax activation for final output layer in both cases as we seek a binary value (for the classifier) and a histogram that sums to one (for the fracture histogram network).

Because SP-WIFF can be run on any arbitrary input, we may generate a training dataset of unlimited size. In this study, we use data from a single year (2009) of a coupled simulation of waves and sea ice (as in the simulation FSD-WAVEv2 in Roach et al. (2019), but here we use a fully-converged wave fracture parameterization instead of a single application of **S1**-**S2**, i.e. SP-WIFF instead of SP-WIFF$_1$). We take 6-hourly wave spectra, sea ice concentration, sea ice thickness, and fracture histograms in both hemispheres, a total of 180 million possible input vectors (most without sea ice). While the coupled wave-ice simulation includes the full distribution of sea ice floe sizes and thicknesses, only mean floe size, mean floe thickness, and sea ice concentration are passed to the wave module, as they are required to compute wave attenuation, and therefore we use these parameters to build NN-WIFF. We prune training data by including only locations where the local sea ice thickness is less than 10 m, sea ice concentration is greater than 0.01, and the significant wave height $H_s = 4\sqrt{E}$ is greater than 0.1 m. These thresholds are also checked before executing SP-WIFF in the version of WIFF released here. This eliminates spurious wave fracture calls in areas of anomalous sea ice conditions or low wave energies. We find a total of 17.9 million sea ice points, of which 5.1 million meet the criterion above.

Thus in total, the full training data set comprises $N = 5.1$ million input vectors of 25 spectral amplitudes, 1 ice thickness, and 1 ice concentration. We assume these include the potential phase space of inputs to the SP-WIFF code, and the potential range of wave energies and sea ice states. These input vectors are identified with a set of output vectors from the "true" SP-WIFF output - in this case 5.1 million 12-category floe probability distributions $A_i dr_i$, of which 2.6 million are non-zero (i.e. SP-WIFF leads to floe fracture). We choose to use the 12 category FSD presented in Roach et al. (2019), however this can readily be altered for different uses. In the WIFF1.0 release, we include code for: (see code and data availability statement)

– Defining a custom training data set using prescribed input spectra and thicknesses.

– Defining a custom output histogram size (i.e. different FSD categories).

– Designing and training the input classification scheme.

– Designing and training the histogram generation scheme.

We also provide data that was used to train networks and produce the results shown here (see Code and Data Availability). We used coupled model output of SP-WIFF for the 12-category floe probability distributions in our training dataset. These can also be generated offline, and we provide offline code (*SP_WIFF_Standalone.m*) for generating output histograms for any arbitrary input that is bit-for-bit the same as the code in SP-WIFF.

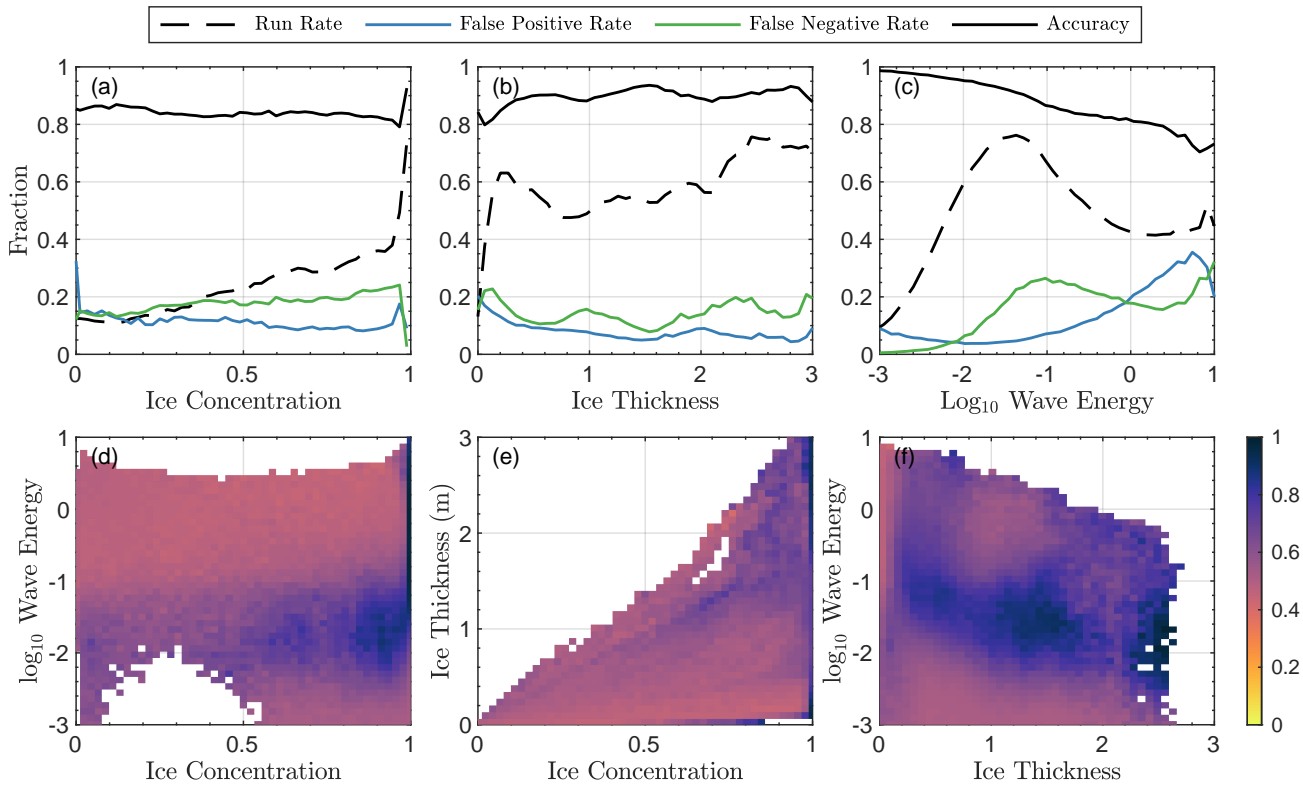

**Figure 1. Performance of Input Classification for NN-WIFF** (a) Percentage of times the classifier runs (run rate, dashed black), calls the histogram network incorrectly (false positive rate, blue), or fails to call the histogram network (false negative rate, green), and overall accuracy (black) for input classifier as a function of sea ice concentration. (b,c) Same as (a) for (b) sea ice thickness and (c), $\log_{10}$ wave energy. (d-f) Heat map of average classifier accuracy as a function of (d) ice concentration and $\log_{10}$ wave energy, (e) ice concentration and ice thickness, and (f) ice thickness and $\log_{10}$ wave energy. Regions of phase space with fewer than 75 input vectors (less than 0.005% of input data) are uncolored/empty.

## 3.1 Input Classification

The histogram $A_i dr_i$ is a collection of numbers that sums to one, and is the target of our parameterization acceleration. This motivates the use of a softmax activation layer (as the output sums to one). Yet when SP-WIFF is executed, sea ice does not always fracture, and a true output histogram could be a vector of zeros. We therefore require a classification layer that determines when a histogram will be produced by NN-WIFF, and when it will not, to reduce the potential for false positives. We found the bulk of potential false positives could be eliminated by introducing the wave energy cutoff of $H_s = 4\sqrt{E} = 0.1$m above, and therefore the classification network is implemented after that cutoff is applied.

The training data for the classifier is the 5.1 million input vectors with corresponding binary values (fracture/no fracture). This is randomly partitioned 70/30 into training ( 3.6 million) and validation ( 1.5 million) datasets, with an even proportion in each set (51%) leading to fracture. The input classifier then takes an input vector and returns a single value, $0 < \chi < 1$, which executes the NN-WIFF scheme if $\chi < \chi_{crit}$. The classification threshold $\chi_{crit}$ is determined by optimizing the error properties of the classifier when evaluated against the validation dataset. We optimize the total error rate, $\epsilon$, where,

$$\epsilon = \frac{\text{False Positives + False Negatives}}{\text{Total Input Vectors}} \tag{5}$$

We found that $\epsilon$ is minimized for $\chi_{crit} = 0.54$ at $\epsilon = 12.5\%$. This error rate corresponds to a false positive rate of 10.6% and a false negative rate of 14.0%.

Fig. 1 shows error characteristics of the classifier evaluated on the validation dataset. In Fig. 1(a-c), we show the relationship between false positive rate (blue) and false negative rate (green) as a function of (Fig. 1a) sea ice concentration, (Fig. 1a) sea ice thickness, and (Fig. 1c) $\log_{10} E$. We also plot the run rate (dashed black line), or the fraction of times NN-WIFF is executed, as well as the overall accuracy (black line), equal to 1-$\epsilon$. Of interest for future model/scientific development is the peak in SP-WIFF calls for wave energies between 0.01 and 0.1 m$^2$ (significant wave heights around 70 cm).

Generally, the input classifier has high accuracy across the range of ice concentrations and thicknesses, with a flat distribution of false positives and false negatives. Lower accuracy occurs at the highest range of wave energies, where both false positives and false negatives are higher than 20%. We plot two-dimensional maps of the overall accuracy in Fig. 1(d) ($\log_{10} E$ vs ice concentration), Fig. 1(e) (ice thickness vs ice concentration), and Fig. 1(f) ($\log_{10} E$ vs ice thickness). Some regions of phase space have classifier accuracies as low as 42%. The classifier accuracy is high in the regions with the highest run rates - high sea ice concentrations, thicknesses, and wave energies between 0.01 and 0.1 m$^2$. As we explore below, these regions represent the largest portion of potential input states, and the 12.5% error in classification does not contribute to statistically significant difference in climate model output. Still, improvements in the input classification may be achievable if desired, for example for higher wave energies, and we include code to re-train the classifier in WIFF1.0.

## 3.2 Neural Network for Generating Fracture Histograms

The fracture histogram network takes a length-27 input vector and outputs a vector $\hat{A}_i dr_i$ which sums to one and approximates the histogram $A_i dr_i$. As a result, sea ice floes will fracture if this input wave and sea ice statistics pass the input classification.

To assess its performance, we use a custom loss function that minimizes the "representative size error", which is the absolute difference in the predicted histogram in each floe size category, weighted by the floe size in that category,

$$RSE = \sum_i \left| \hat{A}_i - A_i \right| r_i \, dr_i, \tag{6}$$

and has units of meters. We choose this error statistic over more standard metrics (root-mean-square error, for example) because mean floe size is an observable quantity (Horvat et al., 2020), and a positive moment of $r$ and therefore less sensitive to errors at small floe sizes or changes in floe size discretization. In the WIFF1.0 release we also provide code for training the neural network using other standard error metrics. Available training data is randomly partitioned into a training dataset of 70% of the input vectors, and a validation dataset of the remaining 30% ($\sim$800,000 input vectors). Training is performed until the validation loss fails to improve over 20 subsequent steps. Indexing training data by $j$, the overall network performance is the average error for the training data,

$$P = \frac{1}{N} \sum_1^N RSE_j \approx 30\text{m}, \tag{7}$$

where $N$ is the total number of inputs used to train the network.

In Figure 2 we display aggregated fit characteristics for the histogram neural net. Figures 2(a-d) display number histograms (red, left axis) of sea ice concentration ($C$), ice thickness ($H$), and $\log_{10}E$ from the validation dataset, as well as of the "true" representative radius $\overline{R}$, where

$$\overline{R} = \sum_i r_i A_i dr_i. \tag{8}$$

from validation dataset input. To evaluate the standardized neural net error, we also compute the standardized size error (SSE) as,

$$SSE = \frac{RSE}{\overline{R}}, \tag{9}$$

which weights errors by the expected mean floe size. Across the entire validation dataset, median $SSE$ is 3.9%.

There is little dependency of SSE on sea ice concentration (2a) or sea ice thickness (2b). Median SSE errors are below 5% across all ice concentrations and thicknesses considered here, with the peak of the interquartile range below 10%. While median errors for both wave energy and $\overline{R}$ also lie below 10% for all input wave energies and true output floe sizes, there are higher SSE values for low wave energies and high $\overline{R}$ values. Large values of $\overline{R}$ can occur for a small number of potential floe fractures, which typically occurs for low wave energies that do not repeatedly fracture the sea ice. In general, the neural network (red line, Fig. 2e) reproduces the mean fracture histogram $\overline{A}_i$ (black line, Fig. 2e) accurately at each floe size category. Median error between $\overline{A}_i dr_i$ and $\hat{\overline{A}}_i dr_i$ is less than 6.2% in all categories, and across all 12 categories averages 3.3%.

To understand what regions of the phase space give rise to error, in Fig. 3(a-c) we plot 2-dimensional heat maps of SSE, evaluated on the validation dataset, for each of the six combinations of ice thickness, concentration, $\overline{R}$, and log wave energy.

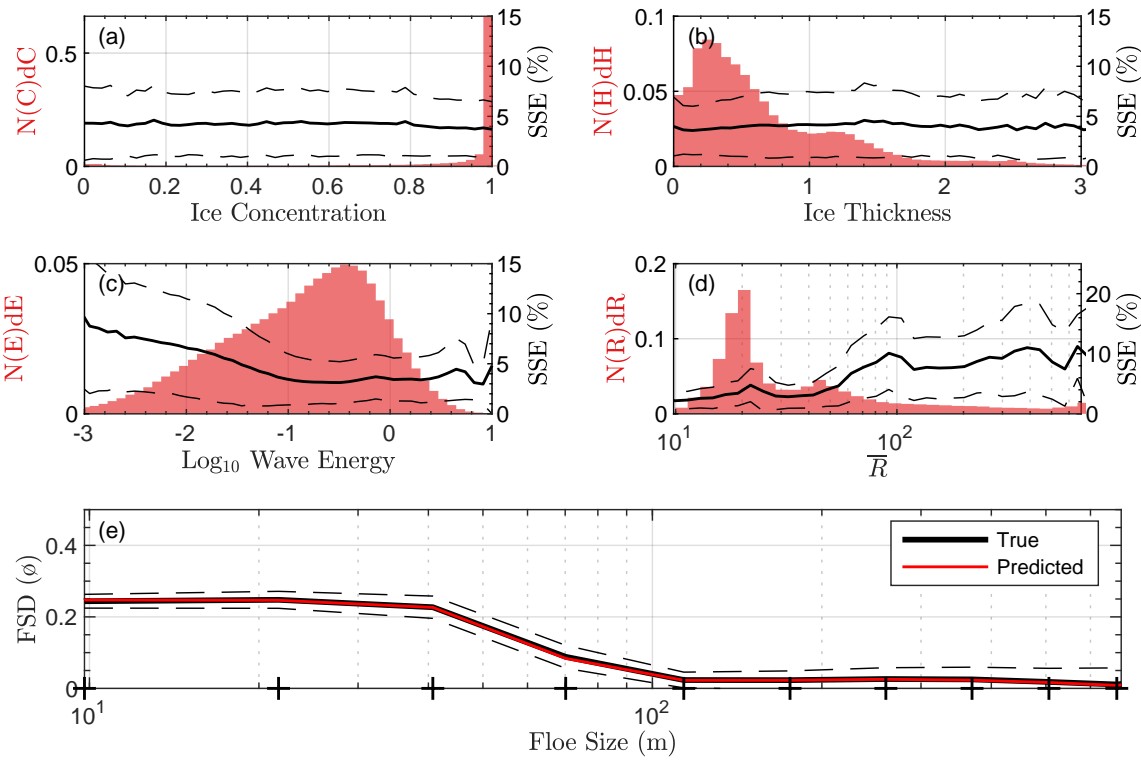

**Figure 2. Performance of NN-WIFF on validation data.** (a, left axis) Histogram of sea ice concentration, $C$, in the validation dataset. (a,right axis) Median standardized size error (SSE, Eq. 9) as a function of $C$ (solid line). Dashed lines are interquartile distance of SSE. (b-d) Same as (a), but for (b) sea ice thickness (c) $\log_{10}$ wave energy, and (d) representative radius ($\overline{R}$, eq. 8) of the "true" output histogram. (e) Mean histogram output $A_i dr_i$ for (black) validation dataset and (red) parameterization output $\hat{F}_i dr_i$. Dashed lines are standard deviation of error between $A_i dr_i$ and $\hat{A}_i dr_i$ in each floe size category. Crosses indicate floe size bin centers.

Colors are shown only for those coordinates with at least 0.005% of the data (40 points), and the whitepoint of the colormap is set for an error of 10%. Blue values are those with SSE < 10%.

Fig. 2 suggests that the input vectors with largest error are those with low input wave energy and high resulting $\overline{R}$. In general, we find similar results when examining these two-dimensional maps. In general, we find low error (SSE<=5%) across the range of input sea ice thickness and concentration (Fig. 3a). However expanding the error map in $\overline{R}$ as a function of thickness (Fig. 3b), concentration (Fig. 3f), and wave energy (Fig. 3c) reveals that the largest part of training error comes from regions with low input sea ice concentration, small thickness, and low wave energy. This combination of inputs gives rise to high output $\overline{R}$ values, which have the highest SSE values. Although locations with low sea ice concentration are not used as WIFF input, we do not presently exclude locations with low sea ice thickness - that these regions produce pathological output suggests they might be isolated or excluded from future versions of NN-WIFF.

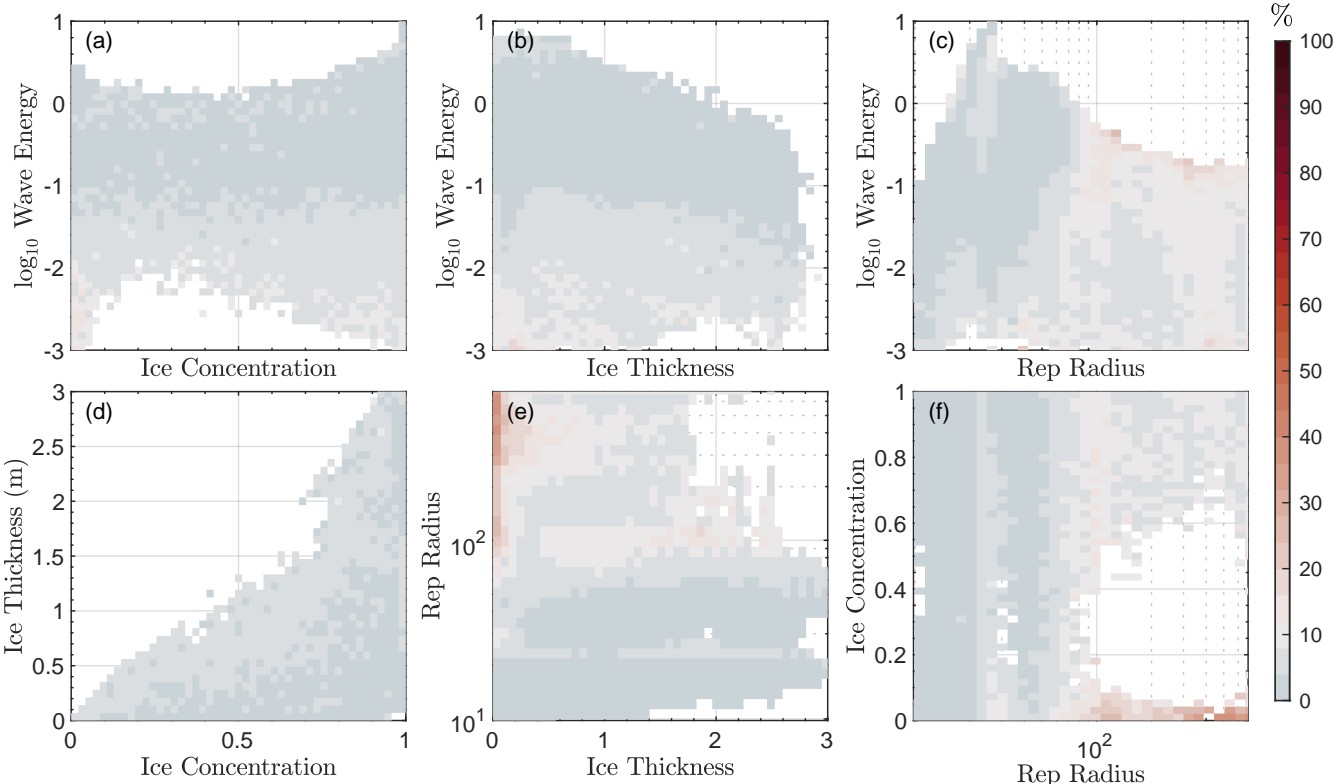

**Figure 3. Examination of Training Errors in NN-WIFF.** (a) 2-D map of median SSE as a function of (x-axis) sea ice concentration and (y-axis) $\log_{10}$E from validation dataset. (b-f) same as (a) but for (b) $H$ and $\log_{10} E$, (c) $\overline{R}$ and $\log_{10} E$, (d) $C$ and $H$, (e) $H$ and $\log_{10} E$, and (f) $\overline{R}$ and $C$. Regions are not colored for less than 0.005% (∼40) points.

## 4  Sea Ice Model Results

We next examine the relationship between versions of the WIFF code when implemented in a free-running sea ice model. Here we use output from the global sea ice model CICE6.1.4 (Hunke et al., 2019) in stand-alone mode forced by the JRA55 atmospheric reanalysis (JRA-55 and Japan Meteorological Agency, 2013) at a nominal $1^o$ spatial resolution. We include time-varying ocean surface wave spectra as sea ice model forcing, taken from output of the CICE-WAVEWATCH III coupled integration described in Roach et al. (2019). The model is run over the year 2005 to provide different boundary conditions and initialization from the 2009 data used as input to the network training.

We examine results from three separate methods for obtaining the output histogram $A_i dr_i$:

1. (SP-WIFF) using the full SP-WIFF model but the converged solution described in Sec. 2

2. (SP-WIFF$_1$) the SP-WIFF model but using a single iteration of steps **S1**-**S2** with fixed phase.

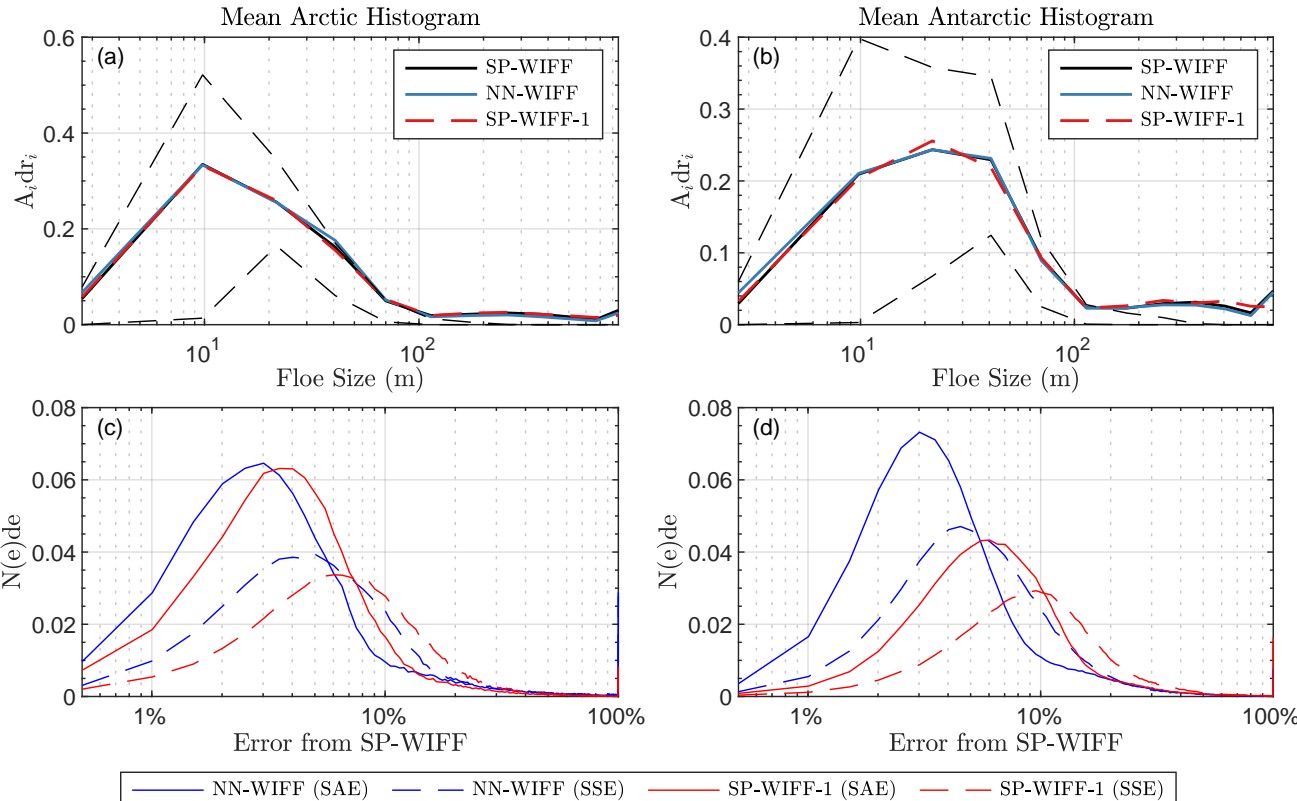

**Figure 4. WIFF error characteristics in a climate model.** (a) Mean Arctic histogram output $A_i dr_i$ in free-running climate simulation using (black) SP-WIFF, (blue) NN-WIFF, and (red dashed) SP-WIFF$_1$. Space/time points included in histogram if all three cause wave fracture. Interquartile range of SP-WIFF outputs given as dashed black lines. (b) Same as (a) for the Antarctic. (c) Histogram of errors between SP-WIFF and (blue) NN-WIFF or (red) SP-WIFF-1 in the Arctic. Solid lines are sum of absolute error (SAE). Dashed lines are standardized size error (SSE). (d) Same as (c), for the Antarctic.

   3. (NN-WIFF) the trained neural network described above.

The histogram output $A_i dr_i$ from CICE is a daily average over 24 hour timesteps. When wave fracture does not occur, $A_i dr_i$ values are set to zero. Thus though $A_i dr_i$ is normalized to one if wave fracture occurs, if a grid cell does not always fracture during one of the averaging timesteps, the CICE history output will not be normalized to one. We choose to include such points in this analysis, renormalizing histogram outputs to sum to one.

Over the course of this year, we find that in the NN-WIFF simulations, there are 686,000 gridpoint-days during which NN-WIFF is executed, for a total of 12.3 million NN-WIFF calls overall. This compares to 746,000 and 14.1 million for SP-WIFF and SP-WIFF-1. The discrepancy between function calls is due to the use of the input classifier, which introduces false negatives as discussed in Sec. 3.1.

In Fig. 4(a,b) we repeat Fig. 2(e) but now examining (a) Arctic and (b) Antarctic averages of $A_i dr_i$ values for each of the three methods. To estimate differences in fracturing, Fig. 4 shows data only from locations and times where WIFF leads to sea ice fracture in all three of the simulations, and where the daily-average wave energy exceeds $0.1$ m$^2$. Dashed lines show the interquartile range of SP-WIFF values. Median values are similar between all three implementations.

We plot histograms of the errors between SP-WIFF and SP-WIFF$_1$ (red) and NN-WIFF (blue) in both hemispheres as Fig. 4(c-d) from the same sample used in producing Fig. 4(a,b). Dashed lines are SSE and solid lines are sum of absolute error (SAE) in each category. In both metrics, and in both hemispheres, NN-WIFF performs as or more accurately as SP-WIFF$_1$ when executed, and with significantly reduced computational cost. In the Arctic, median SSE is 8.9% (NN-WIFF) vs 10.5% (SP-WIFF$_1$), and median SAE is 5.8% (NN-WIFF) vs 5.9% (SP-WIFF$_1$). In the Antarctic, median SSEs are 8.1% (NN-WIFF) vs 13.4% (SP-WIFF$_1$) and median SAEs are 5.52% (NN-WIFF) vs 9.02% (SP-WIFF$_1$).

In Fig. 5(top row) we show monthly-average differences in sea ice concentration between SP-WIFF and NN-WIFF in March and September at each hemisphere. There is little difference between the two, with a maximum global area difference of 47,000 km$^2$ in January (0.3% of total sea ice area). We repeat this analysis on sea ice volume per unit area in Fig. 5(middle row). We again find little difference throughout the year, with a global maximum volume difference of 33 km$^3$ in January (0.12% of total sea ice volume). We also examine the log ratio of SP-WIFF to NN-WIFF mean floe size (representative radius) as the bottom row in Fig. 5. Some areas have larger floe sizes in SP-WIFF compared to NN-WIFF - those where the discrepancy in mean floe sizes is a multiple of two or more, i.e. $|\log_{10}(R_{NN}/R_{SP})| > \log_{10}(2) \approx 0.3$ have a maximum global area of 1100 km$^2$ in April, or 4.8% of total sea ice area. These regions are visible as locations lying at the boundary between the MIZ and sea ice pack, i.e. locations where NN-WIFF does not trigger wave fracture but SP-WIFF does. We evaluate these differences between model integrations using the CICE quality-control check (Roberts et al., 2018). Although there is not bit-for-bit reproducibility, we find that the different implementations of WIFF are not "climate changing" according to the two-stage paired thickness test. This demonstrates that differences in WIFF implementation do not have an emergent effect on sea ice model state, in spite of the impact of wave-ice interactions on sea ice state (Roach et al., 2018b).

NN-WIFF substantially accelerates the parameterization of wave-induced floe fracture. Integrating these simulations on the Cheyenne supercomputer with daily I/O on a 384x320 displaced pole grid, using 1 node and 24 cores, the standard SP-WIFF executes 1 year in approximately 24 hours. The single-iteration version SP-WIFF-1 takes 3h30. The NN-WIFF model takes 1h10 minutes. For comparison, a standard CICE integration with the FSD but no waves takes 56 minutes, and a CICE integration with no FSD and no waves takes 45 minutes. (Note that these runtimes do not include integration of a wave model, as this was run offline and provided as a forcing to the sea ice model.) Thus while execution times will vary with computer architecture and user-specified compiler settings, NN-WIFF significantly reduces the overhead associated with simulating wave-ice fracture.

## 5    Conclusions

Here we have presented WIFF1.0, code for performing wave-induced floe fracture in sea ice models that are coupled to an ocean surface wave model. While the full "super-parameterized" version of WIFF (SP-WIFF) is capable of simulating wave-affected marginal ice zones with active wave-ice interactions, it results in high computational expense which renders it difficult to use in long climate integrations. Instead, we trained a pair of neural networks (NN-WIFF) to accelerate this parameterization using a large set of true climate model input/output pairs, which results in a median "standardized size error" of 3.9% when evaluated on a set of 800,000 input states, and 8.4% in free-running climate model simulations.

Because of the use of a classification layer, NN-WIFF does introduce false positives and negatives - with a classifier accuracy of 87.5%. With an energy threshold for determining whether NN-WIFF is to be called, the classifier network is only called on a reduced subset of sea ice points. The energy threshold eliminates potential calls to NN-WIFF for approximately 72% of total sea ice points, thus the likelihood that NN-WIFF is inappropriately called at any given sea ice point is $0.125 \times 0.28 = 3.5\%$, and the "overall" accuracy of calls to NN-WIFF is 96.5%. When compared to a single-iteration version of SP-WIFF that was previously employed for computational reasons, NN-WIFF produces equal or better error characteristics and is significantly less expensive. Global patterns of sea ice area and volume are statistically indistinguishable between NN-WIFF throughout a year of climate model simulations, with minor differences in representative floe sizes at the boundary between the MIZ and pack ice.

The network has been trained on present-day (2009) sea ice conditions from a single model, and therefore may have less success for different climates where the phase space of wave spectra and ice thicknesses changes. This may present challenges as model projections indicate that the future state of sea ice and wave climate will differ substantially from the present (Casas-Prat and Wang, 2020; Roach et al., 2020; SIMIP Community, 2020, for example), and the typical input sea ice and wave states to NN-WIFF may depart from the phase space examined here. In our training of NN-WIFF, we used a broad range of input model vectors, which sample both geographic and temporal variability in sea ice and wave conditions. While the input dataset spanned only a single year (2009), our assumption is that the future trajectory of ice and wave states lies contained within the geographic and temporal envelope of current model conditions. For example, while thinning Arctic sea ice may lead to larger percentages of the Arctic being affected by waves (Aksenov et al., 2017), interactions between high wave states and thin sea ice are presently simulated along the periphery of the Arctic Ocean and in the Southern Ocean, and so contained within the phase space of the training data examined here.

Still, it is possible that future inputs are not sampled as a part of the present-day conditions we examine here. Existing versions of NN-WIFF can be adaptively re-trained using other, updated climate model output to broaden the phase space of the training dataset, and we provide code for directly producing fracture histograms for any arbitrary wave and sea ice input vectors in WIFF1.0. NN-WIFF model input variables and thresholds may also be adjusted to increase the accuracy of the method, for example the inclusion of a simpler breaking threshold as described by Voermans et al. (2020). We intend in future versions of WIFF to implement a hybrid functionality that permits input vectors that lie outside of the training phase space to be flagged and directly call SP-WIFF to permit better handling of outlier inputs. We note that the phase space of potential

sea ice concentration, thickness, and wave spectral states simulated by the CICE simulations considered here may not reflect the real phase space of these variables, nor that simulated by other climate model configurations. While we assume that CICE-WAVEWATCH III simulations largely mirror this particular set of potential input states, implementation of NN-WIFF in other modeling systems may require re-training to reflect a broader or narrower set of input conditions.

The model of Horvat and Tziperman (2015) was developed under assumptions about how sea ice breaks, namely that it is free-floating and responds instantaneously to curvature in the ocean surface height field. Recent developments in observing wave-induced breakup may point to the use of a single threshold for sea ice floe fracture (Voermans et al., 2020) that could replace the classifier trainer here and improve the representation of floe fracture in SP-WIFF. The aim of developing WIFF1.0 is simply replacing SP-WIFF with an approximating neural network-based code. Future versions of WIFF could be developed in light of new observations (Voermans et al., 2019, e.g.,) and to incorporate more detailed feedbacks between the FSD and wave models. The requirement of passing the full wave spectrum between wave and ice models may be too cumbersome for some current-generation coupling schemes, and it may be possible that fewer model parameters are needed to develop accurate fracture histograms (e.g. the above-mentioned threshold). We provide code in the WIFF1.0 repository for running SP-WIFF in a standalone fashion, collating training data from a climate model, and (re)-training the network, as well as producing the plots used in this paper.

Because of the ease of obtaining training data from climate model output, this parameterization acceleration approach has found a use in many other aspects of climate model simulations (Rasp et al., 2018), for similarly expensive parameterizations, especially those that are not the solution to primitive equations (Pal et al., 2019; Brenowitz et al., 2020). It may have applications in sea ice modeling as well. For example, sea ice rheological parameterizations generally rely on complex implicit solvers run to a specified tolerance, but may, much like SP-WIFF, be able to be decomposed into a series of neural-net "black boxes". This methodology could also be used to accelerate the computation of wave-wave interaction and "source" terms in the WAVEWATCHIII code.

Our primary goal in developing NN-WIFF is to provide a method for accelerating the existing SP-WIFF parameterization, which will permit the implementation of wave-ice fracture parameterizations in climate-scale sea ice models. In general, NN-WIFF allows for a cost-effective implementation of wave-ice fracture in sea ice models with no significant increase in runtime. Because a neural network is a simple set of elementary functions, it can be employed straightforwardly in any climate model that desires to simulate the fracture of sea ice by ocean surface waves.

*Code and data availability.* WIFF1.0 code, code for developing training data, for building and training NN-WIFF, and for implementing into Icepack/CICE6 is released and archived at doi.org/10.5281/zenodo.5793692. Ongoing development of WIFF takes place at https://github.com/chhorvat/WIFF-Model. Model output (from the three 2005-forced WIFF configurations) and training data (from the 2009 SP-WIFF simulation) are available on Zenodo at doi.org/10.5281/zenodo.5106703 and doi.org/10.5281/zenodo.5108636, respectively. Code for CICE6 and Icepack including the implementation of WIFF1.0 is actively developed at https://github.com/lettie-roach/CICE/tree/mlwave and https://github.com/lettie-roach/Icepack/tree/mlwave.

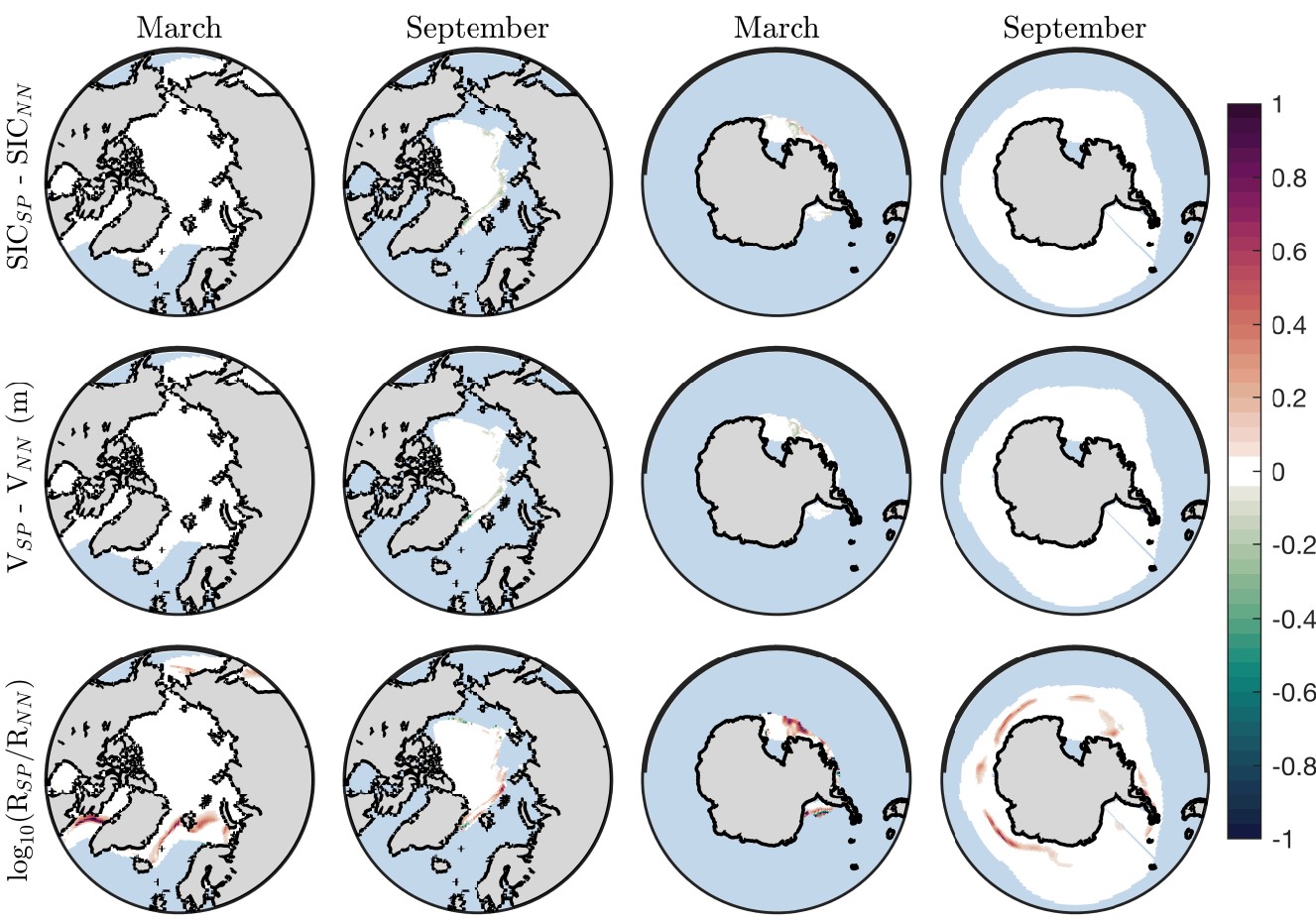

**Figure 5. Implementation of WIFF in a climate model.** (Top row) Difference in monthly sea ice concentration, SIC, between $SP-WIFF$ and $NN-WIFF$ in March and September at each hemisphere. (Middle row) Same as top row, for sea ice volume, V. (Bottom row) Same, for $\log_{10}$ of ratio of representative radius.

*Author contributions.* Both authors designed the study. CH trained and validated the neural networks and wrote the manuscript. LR incorporated NN-WIFF into CICE and performed model runs for training and validation.

340 *Competing interests.* The authors declare no competing interests

*Acknowledgements.* CH was supported by NASA grant 80NSSC20K0959 and by Schmidt Futures – a philanthropic initiative that seeks to improve societal outcomes through the development of emerging science and technologies. CH thanks the National Institute of Water and Atmospheric Research in Wellington, NZ for their hospitality during this work. LR was supported by New Zealand Ministry of Business, Innovation and Employment under Science Investment Contract C01X1914 and US National Science Foundation grant OPP-1643431.

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
