# Peer review of "WIFF1.0: A hybrid machine-learning-based parameterization of Wave-Induced sea-ice Floe Fracture."

_Geoscientific Model Development, 2021_

## Referee Comment (RC1)

**WIFF1.0: A hybrid machine-learning-based parameterization of**
**Wave-Induced sea-ice Floe Fracture**
**Horvat & Roach (2021)**

**General comments**
- This an interesting and well-written paper although I would recommend revisions (somewhere between minor and major) before publication.
- It makes a lot of sense to use a NN to update the FSD depending on the wave field/ice conditions etc, although maybe it should be more flexible (eg having the breaking threshold as an input parameter to the NN).
- One question is if a NN classifier is the best thing to use or can a simpler criterion be applied (eg a threshold in the variance in the strain) that would be
    i. simpler and faster
    ii. physical (as opposed to a black box). I would also mention the paper of Voermans et al (2020) (see https://tc.copernicus.org/articles/14/4265/2020/) who seemed to find such a threshold empirically by collating several observations of sea ice break-up in the presence of waves. This paper is conspicuous in its absence from the bibliography by the way.
    iii. flexible as opposed to being fixed during the training.

**Specific comments**
- P2 l50: Training on model output is a good idea for generating a variety of input wave spectra, but there is potentially other confounding factors – more details could be provided – hopefully it is still only the input and output of the SP-WIFF that is used for training?
- P2 L50: A year should be long enough to give enough variety in conditions – eg from regional, seasonal  differences.
- P4 l95: Perhaps add the RELU acronym for the activation functions as probably not many people know what it stands for (including me)
- P3: S1, S2: why does the algorithm need to converge when there is no ice-to-wave feedback?
- P4 l101, l107-108: you talk about SIC and SIT histograms, but you only input the mean SIC and SIT to the NN? Is the model run with a joint thickness/size distribution or is there just one FSD for all thickness categories?
- Fig 1
    - add "run rate" definition to caption? (I see it is defined later in the text, but it took a while to find it). Can you also define "false positive/negative rate"?
- P6 eq 5: was there a reason for wanting to weight errors in the bins for higher floe sizes more than the bins for smaller ones? Otherwise a simple RMSE might be enough. Another possible metric for comparing PDFs are the Kolmogorov-Smirnov test (this may not be differentiable, but could be used in evaluation). The SSE you've defined is reasonable though.
- Discussion: perhaps an example that could be quite pertinent to the current paper is to speed up calculation of source terms in WW3. In addition, the SP-WIFF could be enhanced by allowing some ice-to-wave feedback and could possibly output a wave source term as well as the FSD.
- For some setups (eg if an interface like OASIS-MCT were used in the wave-ice coupling), using NN-WIFF as provided would require the full wave (frequency) spectrum to be passed to the ice model which is quite costly. Perhaps some kind of dimensionality reduction could be performed to reduce the parameters that had to be passed? This could possibly be done with a NN as well, in order to fit in with the current structure.
- Fig 5, bottom row: 2 orders of magnitude difference in 4.5% of the total sea ice area is not so small; it becomes more significant as a fraction of the MIZ. Since (as you note yourselves) this is on the MIZ-pack boundary it could be worth another look at the classification – I think it is worth another trial run with the classifier replaced by a simpler and perhaps more stable criterion.

**Typos**
- P2 l5: "it overall computation times by an order of magnitude" to "it increases overall computation..."
- p3 l61: equation should possibly be something like this

$$S(\lambda) = \int_0^{2\pi} S(\lambda, \theta) d\theta$$

?

- P11:Integrating these simulations *using on* the Cheyenne supercomputer
- p11: "NN-WIFF reduces the overhead associated with simulating wave-ice fracture without significant added computational cost" to " NN-WIFF reduces the overhead associated with simulating wave-ice fracture"?
- p12 l61-62: "Because of the ease of obtaining training data from climate model output, this parameterization acceleration approach has, and could continue to, find" - "… has found,… to find,"
- p12 l266: paramterization,

---

## Referee Comment (RC2)

**Review of C. Horvat and L. Roach — "WIFF1.0: A hybrid machine-learning-based parameterization of Wave-Induced sea-ice Floe Fracture."**

The authors previously developed a physics-based model (SP-WIFF), which can capture wave-induced sea ice floe fracture and was used as a super-parameterization in the large-scale model CICE. However, including SP-WIFF increased the runtime of CICE by an order of magnitude. Here, the authors develop a neural network model (NN-WIFF), trained on the output of SP-WIFF, that can be used as a computationally efficient alternative to SP-WIFF in large-scale models. It significantly reduced runtime while still capturing the floe fracture patterns produced by SP-WIFF in CICE quite reliably. I found the paper was straightforward and easy to read (apart from section 2). It could be relevant for large-scale sea ice modeling, and I recommend it for publishing after revisions.

My main comment is that I couldn't really understand how SP-WIFF works just from reading the summary in the paper (section 2). I understand that it is explained in detail elsewhere, but it would help to add some more clarifications here so that it would be clearer what the baseline assumptions are — any error introduced by the neural network comes on top of this. Also, I feel that the claim that NN-WIFF is overall superior to physically based SP-WIFF-1, is somewhat unfounded (based on the presented evidence) — for example, under a climate change scenario the neural network could fail as the base state moves outside the parameter range of the training set, while SP-WIFF-1 could still be as accurate as before.

Major comments:

1. section 2: I had trouble understanding the basic assumptions of the model from this section. Here are some of the places where I got confused, but I would appreciate a more detailed discussion in general.

   (a) What is the timescale of Eq. 1 — does it evolve on the same timescale as the large-scale model (e.g. CICE), or is it considered to be infinitely fast compared to large-scale model evolution (i.e. for each time-step of the large-scale model, one finds a steady state distribution $f(r)$)? Do you consider Eq. 1 to be a part of SP-WIFF, or do just the steps S1-S3 fall under it?

   (b) Does this model explicitly ignore the possibility for ice floes to be advected between grid cells? Or is this possibility somehow included independently in the large-scale model?

   (c) At which timescales does the wave spectrum, $S(\lambda)$, evolve compared to the FSD? Is there a feedback with the FSD?

   (d) Why is $F(r,s)ds$ independent of time, $t$, and of the duration $dt$? I feel there is some implicit assumption here that I do not understand. Naively, I would think that the longer one waits and allows the ice to break, the higher fraction of the original floes would end up as small floes?

   (e) Why 10km in step S1? Is that the typical size of a grid cell?

(f) Does the floe size distribution, $f(r)$, enter the fracture algorithm, S1-S3? And, if so, how? It seems to me that it should since $\Omega(r,t)$, as defined at the beginning of this section, should be proportional to $f(r)$.

(g) Where does time enter S1-S3 (to give a time-dependent $\Omega(r,t)$)? Through $f(r)$ and $S(\lambda)$?

(h) Does S3 consist of repeating S1-S2 on the same floe (breaking it additionally with each step), or does each iteration start from a new 10km floe?

(i) I do not understand how steps S1-S3 yield the terms $\Omega$ and $F$ as defined in the opening paragraphs of section 2. In the beginning, $\Omega(r,t)\mathrm{d}r\mathrm{d}t$ is defined as the fraction of the domain for which floes of size between $r$ and $r+\mathrm{d}r$ fracture between times $t$ and $t+\mathrm{d}t$. From Eq. 2, it seems that $\Omega(r,t)$ is the fraction of floes smaller than $r$ that come from fracturing a very large floe (of 10km in size). How are these two definitions equivalent? Moreover, $\Omega$ as defined in the beginning has units of $\mathrm{m}^{-1}\mathrm{s}^{-1}$, whereas $\Omega$ in Eq. 2 is dimensionless. Likewise for $F(r,s)\mathrm{d}s$ — the introduction defines it as the fraction $\Omega(r,t)\mathrm{d}r\mathrm{d}t$ that breaks into floes of size between $s$ and $s+\mathrm{d}s$, while Eq. 3 seems to suggests that it is the ratio of the number of floes of size between $r$ and $r+\mathrm{d}r$ to those smaller than $s$. Again, I cannot reconcile these two definitions. Perhaps a more careful explanation of what $A(r)$ is and how it is related to $\Omega$ and $F$ would help.

(j) Eq. 2: I believe it should be $\Omega(r,t)\mathrm{d}r\mathrm{d}t$ instead of $\Omega(r)$. Also, isn't $\int_0^\infty A(r)\mathrm{d}r = 1$, so that the denominator is unnecessary?

(k) Eq. 3: Shouldn't it be $F(s,r)\mathrm{d}r$ instead of $F(s,r)\mathrm{d}s$?

2. performance of NN-WIFF compared to SP-WIFF-1: A major drawback of neural networks (or any other "black box" method), is that we cannot rely on them in circumstances significantly different than those seen during training. Physically-based models, such as SP-WIFF-1, are not as susceptible to this. So, under a climate change scenario, SP-WIFF-1 could turn out to be a better choice. I feel this point was not really discussed much apart from a couple of sentences in the conclusions. Perhaps a short discussion (if not more investigation) about this would be useful. As a suggestion for future versions of this model (which I don't expect implemented here), it could perhaps be useful to include the possibility to flag data points that fall outside of the parameter range of the training set.

   (a) line 193: The model trained on 2009 data is tested on 2005 — did you compare with other years? Judging from white regions in Figs. 1d-f and 3, there are significant parts of the parameter space that are not visited during training, but perhaps could be visited under different conditions.

   (b) line 213: The difference between NN-WIFF and SP-WIFF-1 errors seems to be quite small, and typically much smaller than the spread of the error (e.g. in the Arctic the difference is 0.1% in the SAE metric). So, saying that NN-WIFF consistently outperforms SP-WIFF-1 seems like an overstatement to me. I would rather say that they are of quite similar accuracy, although NN-WIFF is significantly faster.

   (c) line 250: Again, I am not convinced that NN-WIFF is always more accurate than SP-WIFF-1.

   (d) paragraph of line 255: Perhaps you can expand on this discussion.

3. line 228: "This demonstrates that differences in WIFF implementation do not have an emergent effect on sea ice model state." — It could also be that ice fracture in itself does not have a major impact on the state of sea ice. Have you compared the sea ice state with and without ice fracture (here, or in some previous work)?

Minor comments:

1. line 5: Can you explain what "bitwise reproducible" means?

2. line 17: Does $f(r)$ integrate to 1 or to sea ice concentration?

3. line 22: "Wave-affected sea-ice-covered regions are observed to be several million square kilometers in size in both hemispheres" — is this a significant fraction of the total sea ice area?

4. line 31: What is $E$ exactly — total wave energy in a grid-cell? Or in some other area? $E$ seems to be a normalized so that it has units m$^2$. Could you mention its units and say explicitly how it is normalized?

5. line 45: "it overall computation times by an order of magnitude" —"overall" $\rightarrow$ "increases"?

6. line 61: "unidirectional wave spectrum" — Can you explain what "unidirectional" means here?

7. line 91: What does 100x100 refer to? Hidden layers and nodes? That seems quite large.

8. line 92: "and a second with five hidden layers of 100 nodes each for generating fracture histograms." — add a comma after "each", this way it sounds like each node generates a histogram.

9. Figs. 1-3: Please add units to labels where appropriate (ice thickness, $\bar{R}$, etc.).

10. Fig. 1, caption: Please define "run rate" in the caption. Also, can you say what the white regions in panels d-f are.

11. Fig. 1c: There is a peak run rate at $E$ between 0.01 and 0.1m$^2$. Does this mean that the floes break most at this energy, do you know if there is maybe a physical interpretation for this?

12. Fig. 2, caption: Could you please add references to equations 7 and 8 for $\bar{R}$ and $SSE$ in the caption.

13. Fig. 2e: The way the figure is plotted, it is not clear what the bins are. Distributions look differently if the bins are linear or logarithmic, so it would be useful to show this somehow — e.g. add dots or plot bars.

14. Fig. 3f: Missing an x-label.

15. Fig. 3, caption: What seems white to me are the uncolored points, whereas $SSE = 10\%$ looks grayish-beige. Please either change the color-scale or explain the difference more carefully in the caption.

16. line 245: "Yet the "overall" accuracy of calls to NN-WIFF is 96.5%." — This sentence comes before an explanation of what "overall" means, so it is a bit difficult to read.

17. Fig. 5, top row: Notation $A$ for area could be confused with the distribution $A(r)$.

---

## Author Comment (AC1)

Dear Dr. Robel,

Thank you and the reviewers for their efforts in reading and helping to revise our work. Below all reviewer comments are produced in blue. Our comments are in black, and changes in the manuscript are provided in indented quotes with line numbers.

Christopher Horvat

**Reviewer 1: Timothy Williams**

General comments: This an interesting and well-written paper although I would recommend revisions (somewhere between minor and major) before publication.

Thank you for your assistance in improving the manuscript.

It makes a lot of sense to use a NN to update the FSD depending on the wave field/ice conditions etc, although maybe it should be more flexible (eg having the breaking threshold as an input parameter to the NN).

One question is if a NN classifier is the best thing to use or can a simpler criterion be applied (eg a threshold in the variance in the strain) that would be i. simpler and faster ii. physical (as opposed to a black box). I would also mention the paper of Voermans et al (2020) (see https://tc.copernicus.org/articles/14/4265/2020/) who seemed to find such a threshold empirically by collating several observations of sea ice break-up in the presence of waves. This paper is conspicuous in its absence from the bibliography by the way. iii. flexible as opposed to being fixed during the training.

We have added reference to the Voermans study in the discussion of future work, noting that the development of the physical parameterization for wave fracture (SP-WIFF) was not a target of this particular study (pg 14, line 311) although it should be incorporated into future iterations:

> Recent developments in observing wave-induced breakup may point to the use of a single threshold for sea ice floe fracture (*Voermans et al.*, 2020) that could replace the classifier trainer here and improve the representation of floe fracture in SP-WIFF. The aim of developing WIFF1.0 is simply replacing SP-WIFF with an approximating neural network-based code. Future versions of WIFF could be developed in light of new observations (*Voermans et al.*, 2019, e.g.,) and to incorporate more detailed feedbacks between the FSD and wave models.

We also discuss when introducing the classifier layer (pg 4, line 114),

> We introduced a classifier layer as SP-WIFF frequently returns un-fractured sea ice in low-wave regimes, and we wish to train and run the histogram-generating network only when the sea ice will fracture. Recent observations of universal threshold behavior for sea ice break-up (*Voermans et al.*, 2020) may provide the opportunity in future work to replace this classification layer in future work with a simple physically-based threshold for when the sea ice fractures.

Specific comments P2 l50: Training on model output is a good idea for generating a variety of input

wave spectra, but there is potentially other confounding factors – more details could be provided – hopefully it is still only the input and output of the SP-WIFF that is used for training?

Indeed it is. We add slightly more context and point to where the training data is provided (pg 2, line 55),

> The model is trained using 5.1 million input and output vectors taken from coupled CICE-WAVEWATCH 3 simulations (see Sec.XX).

We also added substantial discussion of challenges implied by this approach (see response to reviewer 2).

P2 L50: A year should be long enough to give enough variety in conditions – eg from regional, seasonal differences.

We agree - although we discuss potential limitations in the discussion (pg 13, line 287)

> The network has been trained on present-day (2009) sea ice conditions from a single model, and therefore may have less success for different climates where the phase space of wave spectra and ice thicknesses changes. This may present challenges as model projections indicate that the future state of sea ice and wave climate will differ substantially from the present (*Casas-Prat and Wang*, 2020; *Roach et al.*, 2020; *SIMIP Community*, 2020, for example), and the typical input sea ice and wave states to NN-WIFF may depart from the phase space examined here. In our training of NN-WIFF, we used as broad a range of input model vectors, which sample both geographic and temporal variability in sea ice and wave conditions. While the input dataset spanned only a single year (2009), our assumption is that the future trajectory of ice and wave states lies contained within the geographic and temporal envelope of current model conditions. For example, while thinning Arctic sea ice may lead to larger percentages of the Arctic being affected by waves (*Aksenov et al.*, 2017), interactions between high wave states and thin sea ice are presently simulated along the periphery of the Arctic Ocean and in the Southern Ocean, and so contained within the phase space of the training data examined here.

P4 l95: Perhaps add the RELU acronym for the activation functions as probably not many people know what it stands for (including me)

We add now (pg 5, line 121)

> Rectified linear unit (RELU) activation functions are used for each of the hidden neurons,

P3: S1, S2: why does the algorithm need to converge when there is no ice-to-wave feedback?

The convergence is to a steady-state distribution of fractures, since it has a stochastic element, not a response to a feedback. We clarify (pg 4, line 103),

> Note that this convergence is related to the stochastic histogram generation code, which converges to a steady-state distribution of fractures, and is not related to feedbacks from

the waves to the ice.

P4 l101, l107-108: you talk about SIC and SIT histograms, but you only input the mean SIC and SIT to the NN? Is the model run with a joint thickness/size distribution or is there just one FSD for all thickness categories?

The model is run with the joint FSTD, but we send only the thickness and concentration to the NN. We explain this (pg 5, line 128),

> While the coupled wave-ice simulation includes the full distribution of sea ice floe sizes and thicknesses, only mean floe size, mean floe thickness, and sea ice concentration are passed to the wave module, as they are required to compute wave attenuation, and therefore we use these parameters to build NN-WIFF.

Fig 1 ∘ add "run rate" definition to caption? (I see it is defined later in the text, but it took a while to find it). Can you also define "false positive/negative rate"?

We define this terms in the caption now (Fig 1)

> (a) Percentage of times the classifier runs (run rate, dashed black), calls the histogram network incorrectly (false positive rate, blue), or fails to call the histogram network (false negative rate, green),

P6 eq 5: was there a reason for wanting to weight errors in the bins for higher floe sizes more than the bins for smaller ones? Otherwise a simple RMSE might be enough. Another possible metric for comparing PDFs are the Kolmogorov-Smirnov test (this may not be differentiable, but could be used in evaluation). The SSE you've defined is reasonable though.

We explain this choice in (pg 7, line 185),

> We choose this error statistic over more standard metrics (root-mean-square error, for example) because mean floe size is an observable quantity *Horvat et al.* (2020), and a positive moment of $r$ and therefore less sensitive to errors at small floe sizes or changes in floe size discretization. In the WIFF1.0 release we also provide code for training the neural network using other standard error metrics.

Discussion: perhaps an example that could be quite pertinent to the current paper is to speed up calculation of source terms in WW3. In addition, the SP-WIFF could be enhanced by allowing some ice-to-wave feedback and could possibly output a wave source term as well as the FSD.

Indeed this is a good point. We add some discussion (pg 14, line 313).

> The aim of developing WIFF1.0 is simply replacing SP-WIFF with an approximating neural network-based code. Future versions of WIFF could be developed in light of new observations (*Voermans et al.*, 2019, e.g.,) and to incorporate more detailed feedbacks between the FSD and wave models.

and (pg 14, line 326),

This methodology could also be used to accelerate the computation of wave-wave interaction and "source" terms in the WaveWatchIII code.

For some setups (eg if an interface like OASIS-MCT were used in the wave-ice coupling), using NN-WIFF as provided would require the full wave (frequency) spectrum to be passed to the ice model which is quite costly. Perhaps some kind of dimensionality reduction could be performed to reduce the parameters that had to be passed? This could possibly be done with a NN as well, in order to fit in with the current structure.

We discuss this, and how we can reduce the input space (pg 14, line 316)

The requirement of passing the full wave spectrum between wave and ice models may be too cumbersome for some current-generation coupling schemes, and it may be possible that fewer model parameters are needed to develop accurate fracture histograms (e.g. the above-mentioned threshold).

Fig 5, bottom row: 2 orders of magnitude difference in 4.5% of the total sea ice area is not so small; it becomes more significant as a fraction of the MIZ. Since (as you note yourselves) this is on the MIZ-pack boundary it could be worth another look at the classification – I think it is worth another trial run with the classifier replaced by a simpler and perhaps more stable criterion.

We re-wrote this section as the difference was not two orders of magnitude but a factor of two (pg 12, line 255),

Some areas have larger floe sizes in SP-WIFF compared to NN-WIFF - those where the discrepancy in mean floe sizes is a multiple of two or more, i.e. $|\log_{10}(R_{NN}/R_{SP})| > \log_{10}(2) \approx 0.3$ have a maximum global area of 1100 km$^2$ in April, or 4.8% of total sea ice area.

We agree that the classifier is an imperfect choice, and discuss how we can move to a different model for this set of equations in a future version (pg 4, line 114),

We introduced a classifier layer as SP-WIFF frequently returns un-fractured sea ice in low-wave regimes, and we wish to train and run the histogram-generating network only when the sea ice will fracture. Recent observations of universal threshold behavior for sea ice break-up (*Voermans et al.*, 2020) may provide the opportunity in future work to replace this classification layer with a simple physically-based threshold for when the sea ice fractures.

Typos: P2 l5: "it overall computation times by an order of magnitude" to "it increases overall computation..."

Fixed!

p3 l61: equation should possibly be something like this ?

We broke out this equation as (pg 3, line 64),

Consider a region corresponding to a climate model grid cell where ocean surface wave

energetics are described by a discrete uni-directional wave spectrum:

$$S(\lambda_i)d\lambda_i = \int\limits_0^{2\pi} S(\lambda_i, \Theta)d\lambda_i d\Theta. \tag{1}$$

P11:Integrating these simulations using on the Cheyenne supercomputer

Fixed!

p11: "NN-WIFF reduces the overhead associated with simulating wave-ice fracture without significant added computational cost" to " NN-WIFF reduces the overhead associated with simulating wave-ice fracture"?

We altered to (pg 12, line 268),

> Thus while execution times will vary with computer architecture and user-specified compiler settings, NN-WIFF significantly reduces the overhead associated with simulating wave-ice fracture.

p12 l61-62: "Because of the ease of obtaining training data from climate model output, this parameterization acceleration approach has, and could continue to, find" - "... has found,... to find,"

We re-wrote this section (pg 14, line 321),

> Because of the ease of obtaining training data from climate model output, this parameterization acceleration approach has found a use in many other aspects of climate model simulations (*Rasp et al.*, 2018), for similarly expensive parameterizations, especially those that are not the solution to primitive equations (*Pal et al.*, 2019; *Brenowitz et al.*, 2020). It may have applications in sea ice modeling as well. For example, sea ice rheological parameterizations generally rely on complex implicit solvers run to a specified tolerance, but may, much like SP-WIFF, be able to be decomposed into a series of neural-net "black boxes". This methodology could also be used to accelerate the computation of wave-wave interaction and "source" terms in the WaveWatchIII code.

p12 l266: paramterization

Fixed!

**Reviewer 2: Predrag Popovic**

The authors previously developed a physics-based model (SP-WIFF), which can capture wave-induced sea ice floe fracture and was used as a super-parameterization in the large-scale model CICE. However, including SP-WIFF increased the runtime of CICE by an order of magnitude. Here, the authors develop a neural network model (NN-WIFF), trained on the output of SP-WIFF, that can be used as a computationally efficient alternative to SP-WIFF in large-scale models. It significantly reduced runtime while still capturing the floe fracture patterns produced by SP-WIFF in CICE quite reliably. I found the paper was straightforward and easy to read (apart from section 2). It could be relevant for large-scale sea ice modeling, and I recommend it for publishing after revisions.

Thank you for your assistance in improving the manuscript.

My main comment is that I couldn't really understand how SP-WIFF works just from reading the summary in the paper (section 2). I understand that it is explained in detail elsewhere, but it would help to add some more clarifications here so that it would be clearer what the baseline assumptions are — any error introduced by the neural network comes on top of this. Also, I feel that the claim that NN-WIFF is overall superior to physically based SP-WIFF-1, is somewhat unfounded (based on the presented evidence) — for example, under a climate change scenario the neural network could fail as the base state moves outside the parameter range of the training set, while SP-WIFF-1 could still be as accurate as before.

Thank you for these main comments. We have lengthened the description of the core SP-WIFF code, and included further discussion about the drawbacks of using a NN-based scheme following the detailed comments below.

Major comments:

1. section 2: I had trouble understanding the basic assumptions of the model from this section. Here are some of the places where I got confused, but I would appreciate a more detailed discussion in general.
   Thanks, we reply to all in line below. We agree that this manuscript requires more description of the original scheme and have added this in Section 2 throughout.

   (a) What is the timescale of Eq. 1 — does it evolve on the same timescale as the large-scale model (e.g. CICE), or is it considered to be infinitely fast compared to large-scale model evolution (i.e. for each time-step of the large-scale model, one finds a steady state distribution f(r))? Do you consider Eq. 1 to be a part of SP-WIFF, or do just the steps S1-S3 fall under it?
   We re-wrote this section to more accurately represent what Eq. 1 entails (pg 3, line 64),

   > Consider a region corresponding to a climate model grid cell where ocean surface wave energetics are described by a discrete uni-directional wave spectrum:

   $$S(\lambda_i)d\lambda_i = \int\limits_0^{2\pi} S(\lambda_i, \Theta)d\lambda_i d\Theta. \qquad (2)$$

   > Considering only those floes with horizontal size between $r$ and $r+dr$, a fraction of the domain $\frac{f(r)}{\tau}\Omega(r,t)\,dr\,dt$ (unitless) is broken by ocean surface waves over a

period $dt$. The parameter $\tau$ is a prescribed timescale over which the floe fracture takes place. In the *Roach et al.* (2018a) implementation, $\tau$ is determined via an adapting timestepping algorithm to satisfy the CFL criteria for fast-propagating waves and small-area FSD categories, respectively (see *Horvat and Tziperman* (2017), Appendix A for further details).

And we discuss timstepping of equations like Eq.1 at (pg 3, line 77),

External forcing terms are computed at each model timestep, but the FSD $f(r,t)$ is prognostically evolved using the above-referenced adaptive timestepping scheme. Wave-ice coupling is performed at each sea ice model grid step, allowing for feedbacks between the FSD and, for example, floe-size-dependent wave attenuation schemes (*Meylan et al.*, 2021).

and specifically discuss Eq. 1 as not being SP-WIFF (pg 3, line 83),

As Eq. 4 is a generic tendency equation for fractured sea ice, SP-WIFF, then, refers to a parameterization of both $\Omega$ and $F$, which in each ice thickness category are evaluated as follows:

(b) Does this model explicitly ignore the possibility for ice floes to be advected between grid cells? Or is this possibility somehow included independently in the large-scale model?

The wave fracture code is one tendency of several for FSD evolution - we are now more explicit (pg 3, line 77),

Equation 4 is one tendency term in evolution of the FSD, which responds to multiple external forcings (e.g., thermodynamic growth/melting and advection) as described in *Horvat and Tziperman* (2015). External forcing terms are computed at each model timestep, but the FSD $f(r,t)$ is prognostically evolved using the above-referenced adaptive timestepping scheme.

(c) At which timescales does the wave spectrum, $S(\lambda)$, evolve compared to the FSD? Is there a feedback with the FSD?

We discuss the coupling between the wave model and sea ice model (pg 3, line 78)

External forcing terms are computed at each model timestep, but the FSD $f(r,t)$ is prognostically evolved using the above-referenced adaptive timestepping scheme.

and feedbacks between the two (pg 3, line 79),

Wave-ice coupling is performed at each sea ice model grid step, allowing for feedbacks between the FSD and, for example, floe-size-dependent wave attenuation schemes (*Meylan et al.*, 2021).

(d) Why is F(r, s)ds independent of time, t, and of the duration dt? I feel there is some implicit assumption here that I do not understand. Naively, I would think that the longer one waits and allows the ice to break, the higher fraction of the original floes would end up as small floes?

Yes, this is a good catch! We update the text to include the missing time scale (pg 3, line 72),

Generically, the time rate of change of area of floes of size $r$ due to fracture by

ocean surface waves is,

$$\frac{\partial f(r,t)}{\partial t} = \frac{1}{\tau}\left[-f(r,t)\Omega(r,t) + \int_r^\infty \Omega(s,t)f(s,t)F(s,r,t)\,ds\right]. \qquad (3)$$

Note that for ease of interpretation, notation in Eq. 4 differs from the analogous equation set in *Horvat and Tziperman* (2015), Eq.s 19-23. Equation 4 is one tendency term in evolution of the FSD, which responds to multiple external forcings (e.g., thermodynamic growth/melting and advection) as described in *Horvat and Tziperman* (2015).

(e) Why 10km in step S1? Is that the typical size of a grid cell?
This was an arbitrary choice, we add (pg 3, line 85),

The discrete unidirectional wave spectrum $S(\lambda)d\lambda$ is converted to a 1-dimensional ice strain field $\eta(x)$ of (arbitrarily chosen) length 10 km, (see (*Horvat and Tziperman*, 2015, eq. 20-21))

(f) Does the floe size distribution, f(r), enter the fracture algorithm, S1-S3? And, if so, how? It seems to me that it should since $\Omega(r, t)$, as defined at the beginning of this section, should be proportional to f(r).
It does, but this was buried in the definition of $\Omega$. We altered notation to be more clear (pg 3, line 72),

Generically, the time rate of change of area of floes of size $r$ due to fracture by ocean surface waves is,

$$\frac{\partial f(r,t)}{\partial t} = \frac{1}{\tau}\left[-f(r,t)\Omega(r,t) + \int_r^\infty \Omega(s,t)f(s,t)F(s,r,t)\,ds\right]. \qquad (4)$$

Note that for ease of interpretation, notation in Eq. 4 differs from the analogous equation set in *Horvat and Tziperman* (2015), Eq.s 19-23.

(g) Where does time enter S1-S3 (to give a time-dependent $\Omega(r, t)$)? Through f(r) and $S(\lambda)$?
See above, $\Omega$ in *Horvat and Tziperman* (2015) did include a time component, which we broke out in translating to this manuscript.

(h) Does S3 consist of repeating S1-S2 on the same floe (breaking it additionally with each step), or does each iteration start from a new 10km floe?
We clarify (pg 4, line 93),

Each wave field and corresponding fracture length collection (steps **S1** and **S2**) is performed independently, i.e. on an unbroken floe of length 10km.

(i) I do not understand how steps S1-S3 yield the terms $\Omega$ and F as defined in the opening paragraphs of section 2. In the beginning, $\Omega(r, t)$drdt is defined as the fraction of the domain for which floes of size between r and r + dr fracture between times t and t+ dt. From Eq. 2, it seems that $\Omega(r, t)$ is the fraction of floes smaller than r that come from fracturing a very large floe (of 10km in size). How are these two definitions equivalent? Moreover, $\Omega$ as defined in the beginning has units of $m^{-1}s^{-1}$, whereas $\Omega$ in Eq. 2 is dimensionless. Likewise for F(r, s)ds — the introduction defines it as the fraction $\Omega$(r,

t)drdt that breaks into floes of size between s and s+ ds, while Eq. 3 seems to suggests that it is the ratio of the number of floes of size between r and r + dr to those smaller than s. Again, I cannot reconcile these two definitions. Perhaps a more careful explanation of what A(r) is and how it is related to $\Omega$ and F would help.

Thanks for highlighting our hard-to-parse description. Above we revised how $\Omega$ was defined to incorporate the time-varying component. We also now describe in more detail the derivation and description of $\Omega$ and $F$ from the histogram (pg 4, line 94),

We next use the histogram $A(r)dr$ to compute $\Omega$ and $F$. First,

$$\Omega(r) = \int_0^r A(s)ds / \int_0^\infty A(s)ds \tag{5}$$

$\Omega(r,t)\,dr$ is equal to the length-weighted fraction of all fracture lengths smaller than $r$, which assuming a random horizontal distribution of floes is also equal to the fraction of floes of size $r$ that will be broken. The resulting distribution of floe sizes is determined by the histogram itself,

$$F(s,r)ds = \begin{cases} A(r)dr / \int_0^s A(r)dr & \text{if } s \geq r, \\ 0 & \text{if } s < r. \end{cases} \tag{6}$$

Thus $F(s,r)$ is the probability distribution of floe sizes from a broken floe of size $s$, and is normalized to one for each $s$.

(j) Eq. 2: I believe it should be $\Omega(r, t)drdt$ instead of $\Omega(r)$. Also, isn't R $\infty$ 0 A(r)dr = 1, so that the denominator is unnecessary?

Thanks for catching this. In our revised description, breaking out the FSD and time-dependent parts of $\Omega$, it retains this form. The denominator indeed is zero in this way of describing the code, and has been removed.

(k) Eq. 3: Shouldn't it be F(s, r)dr instead of F(s, r)ds?

Please see the updated equation and description, which we believe is accurate (pg 4, line 97),

The resulting distribution of floe sizes is determined by the histogram itself,

$$F(r,s,t)ds = \begin{cases} A(s,t)ds / \int_0^r A(l,t)dl = A(s,t)ds/\Omega(r,t) & \text{if } r \geq s, \\ 0 & \text{if } r < s. \end{cases} \tag{7}$$

2. performance of NN-WIFF compared to SP-WIFF-1: A major drawback of neural networks (or any other "black box" method), is that we cannot rely on them in circumstances significantly different than those seen during training. Physically-based models, such as SP-WIFF-1, are not as susceptible to this. So, under a climate change scenario, SP-WIFF-1 could turn out to be a better choice. I feel this point was not really discussed much apart from a couple of sentences in the conclusions. Perhaps a short discussion (if not more investigation) about this would be useful. As a suggestion for future versions of this model (which I don't expect implemented here), it could perhaps be useful to include the possibility to flag data points that fall outside of the parameter range of the training set.

Thanks - this is a good point, and we address the question of outliers in expanded discussion paragraphs (pg 13, line 287):

> In our training of NN-WIFF, we used a broad range of input model vectors, which sample both geographic and temporal variability in sea ice and wave conditions. While the input dataset spanned only a single year (2009), our assumption is that the future trajectory of ice and wave states lies contained within the geographic and temporal envelope of current model conditions. For example, while thinning Arctic sea ice may lead to larger percentages of the Arctic being affected by waves (*Aksenov et al.*, 2017), interactions between high wave states and thin sea ice are presently simulated along the periphery of the Arctic Ocean and in the Southern Ocean, and so contained within the phase space of the training data examined here. Still, it is possible that future inputs are not sampled as a part of the present-day conditions we examine here. Existing versions of NN-WIFF can be adaptively re-trained using other, updated climate model output to broaden the phase space of the training dataset, and we provide code for directly producing fracture histograms for any arbitrary wave and sea ice input vectors in WIFF1.0. NN-WIFF model input variables and thresholds may also be adjusted to increase the accuracy of the method, for example the inclusion of a simpler breaking threshold as described by *Voermans et al.* (2020). We intend in future versions of WIFF to implement a hybrid functionality that permits input vectors that lie outside of the training phase space to be flagged and directly call SP-WIFF to permit better handling of outlier inputs. We note that the phase space of potential sea ice concentration, thickness, and wave spectral states simulated by the CICE simulations considered here may not reflect the real phase space of these variables, nor that simulated by other climate model configurations. While we assume that CICE-Wavewatch III simulations largely mirror this particular set of potential input states, implementation of NN-WIFF in other modeling systems may require re-training to reflect a broader or narrower set of input conditions.

(a) line 193: The model trained on 2009 data is tested on 2005 — did you compare with other years? Judging from white regions in Figs. 1d-f and 3, there are significant parts of the parameter space that are not visited during training, but perhaps could be visited under different conditions.
This is a good point. We do assume that CICE will have relatively stationary sea ice statistics (see above for more discussion on this point). This also raises an important part about other modeling systems who might implement this parameterization (pg 13, line 306):

> While we assume that CICE-Wavewatch III simulations largely mirror this particular set of potential input states, implementation of NN-WIFF in other modeling systems may require re-training to reflect a broader or narrower set of input conditions.

(b) line 213: The difference between NN-WIFF and SP-WIFF-1 errors seems to be quite small, and typically much smaller than the spread of the error (e.g. in the Arctic the difference is 0.1% in the SAE metric). So, saying that NN-WIFF consistently outperforms SP-WIFF-1 seems like an overstatement to me. I would rather say that they are of quite similar accuracy, although NN-WIFF is significantly faster.

We now state (pg 12, line 246),

> In both metrics, and in both hemispheres, NN-WIFF performs as or more accurately as SP-WIFF$_1$ when executed, and with significantly reduced computational cost.

(c) line 250: Again, I am not convinced that NN-WIFF is always more accurate than SP-WIFF-1.

We now say (pg 13, line 282),

> When compared to a single-iteration version of SP-WIFF that was previously employed for computational reasons, NN-WIFF produces equal or better error characteristics and is significantly less expensive.

(d) paragraph of line 255: Perhaps you can expand on this discussion.

Please see above where this section has been largely re-written!

3. line 228: "This demonstrates that differences in WIFF implementation do not have an emergent effect on sea ice model state." — It could also be that ice fracture in itself does not have a major impact on the state of sea ice. Have you compared the sea ice state with and without ice fracture (here, or in some previous work)?

Indeed, we now add (pg 12, line 261),

> This demonstrates that differences in WIFF implementation do not have an emergent effect on sea ice model state, in spite of the known significant impact of wave-ice interactions on sea ice state *Roach et al.* (2018b); *Bateson et al.* (2020).

Minor comments:

1. line 5: Can you explain what "bitwise reproducible" means?

We add (pg 2, line 48)

> As it is stochastic SP-WIFF is not bitwise reproducible: two identically initialized and forced simulations using SP-WIFF will not produce identical output up to the level of machine precision, which is often a necessary feature for climate model development.

2. line 17: Does f(r) integrate to 1 or to sea ice concentration?

We add (pg 1, line 18),

> The integral of $f(r)$ over all positive $r$ is equal to the sea ice concentration, $c$.

3. line 22: "Wave-affected sea-ice-covered regions are observed to be several million square kilometers in size in both hemispheres" — is this a significant fraction of the total sea ice area?

We now state (pg 1, line 22),

> Wave-affected sea-ice-covered regions are observed to be several million square kilometers in size in both hemispheres, impacting up to half of the sea ice cover depending on the season and hemisphere (*Horvat et al.*, 2020).

4. line 31: What is E exactly — total wave energy in a grid-cell? Or in some other area? E seems to be a normalized so that it has units m2. Could you mention its units and say explicitly how it is normalized?

We add (pg 2, line 32),

In a climate model, coupled wave-ice feedbacks are related to two sub-grid-scale distributions: the FSD, $f(r)$, and the ocean surface wave spectrum, $S(\lambda)$, where $\lambda$ is the wavelength and $\int S(\lambda)d\lambda = E$ (units m$^2$) is the wave energy per square meter, with $E = 4\sqrt{H_s}$ for $H_s$ the significant wave height *Michel* (1968).

5. line 45: "it overall computation times by an order of magnitude" —"overall" → "increases"?
   Fixed!

6. line 61: "unidirectional wave spectrum" — Can you explain what "unidirectional" means here?
   We changed to "one-dimensional"

7. line 91: What does 100x100 refer to? Hidden layers and nodes? That seems quite large.
   We changed to "100 node by 100 node".

8. line 92: "and a second with five hidden layers of 100 nodes each for generating fracture histograms." — add a comma after "each", this way it sounds like each node generates a histogram.
   Fixed!

9. Figs. 1-3: Please add units to labels where appropriate (ice thickness, R$^-$, etc.).
   Fixed!

10. Fig. 1, caption: Please define "run rate" in the caption. Also, can you say what the white regions in panels d-f are.
    Caption now reads (Fig 1):

    (a) Percentage of times the classifier runs (run rate, dashed black), calls the histogram network incorrectly (false positive rate, blue), or fails to call the histogram network (false negative rate, green), and overall accuracy (black) for input classifier as a function of sea ice concentration. (b,c) Same as (a) for (b) sea ice thickness and (c), $\log_{10}$wave energy. (d-f) Heat map of average classifier accuracy as a function of (d) ice concentration and $\log_{10}$wave energy, (e) ice concentration and ice thickness, and (f) ice thickness and $\log_{10}$wave energy. Regions of phase space with fewer than 75 input vectors (less than 0.005% of input data) are uncolored/empty.

11. Fig. 1c: There is a peak run rate at E between 0.01 and 0.1m2 . Does this mean that the floes break most at this energy, do you know if there is maybe a physical interpretation for this?
    We do not have a particularly good explanation for this: we comment in the manuscript (pg 7, line 168),

    Of interest for future model/scientific development is the peak in SP-WIFF calls for wave energies between 0.01 and 0.1 m$^2$ (significant wave heights around 70 cm).

12. Fig. 2, caption: Could you please add references to equations 7 and 8 for R$^-$ and SSE in the caption.
    Done!

13. Fig. 2e: The way the figure is plotted, it is not clear what the bins are. Distributions look differently if the bins are linear or logarithmic, so it would be useful to show this somehow — e.g. add dots or plot bars.
    We add crosses to show bin locations, adding to the caption (Fig 2)

Crosses indicate floe size bin centers.

14. Fig. 3f: Missing an x-label.
   Fixed!

15. Fig. 3, caption: What seems white to me are the uncolored points, whereas SSE = 10% looks grayishbeige. Please either change the color-scale or explain the difference more carefully in the caption.
   We deleted the reference to a "white point" as it was an uncaught typo from using a different color map.

16. line 245: "Yet the "overall" accuracy of calls to NN-WIFF is 96.5%." — This sentence comes before an explanation of what "overall" means, so it is a bit difficult to read.
   We rewrote this section (pg 12, line 278), )

   Because of the use of a classification layer, NN-WIFF does introduce false positives and negatives - with a classifier accuracy of 87.5%. Because we now employ an energy threshold for determining whether NN-WIFF would be called, the classifier network is only called on a reduced subset of sea ice points. The energy threshold eliminates potential calls to NN-WIFF for approximately 72% of total sea ice points, thus the likelihood that NN-WIFF is inappropriately called at any given sea ice point is $0.125 \times 0.28 = 3.5\%$, and the "overall" accuracy of calls to NN-WIFF is 96.5%.

17. Fig. 5, top row: Notation A for area could be confused with the distribution A(r).
   We now change to "SIC".